# Molecular Investigation of Zoonotic Intestinal Protozoa in Pet Dogs and Cats in Yunnan Province, Southwestern China

**DOI:** 10.3390/pathogens10091107

**Published:** 2021-08-30

**Authors:** Yu-Gui Wang, Yang Zou, Ze-Zhong Yu, Dan Chen, Bin-Ze Gui, Jian-Fa Yang, Xing-Quan Zhu, Guo-Hua Liu, Feng-Cai Zou

**Affiliations:** 1Key Laboratory of Veterinary Public Health of Yunnan Province, College of Veterinary Medicine, Yunnan Agricultural University, Kunming 650201, China; wyg18894032073@outlook.com (Y.-G.W.); 2002012@ynau.edu.cn (J.-F.Y.); xingquanzhu1@hotmail.com (X.-Q.Z.); 2State Key Laboratory of Veterinary Etiological Biology, Key Laboratory of Veterinary Parasitology of Gansu Province, Lanzhou Veterinary Research Institute, Chinese Academy of Agricultural Sciences, Lanzhou 730046, China; 3Hunan Provincial Key Laboratory of Protein Engineering in Animal Vaccines, College of Veterinary Medicine, Hunan Agricultural University, Changsha 410128, China; GBZ296936@163.com (B.-Z.G.); liuguohua@hunau.edu.cn (G.-H.L.); 4Department of Animal Science, Yuxi Agricultural Vocation Technical College, Yuxi 653106, China; yzznzy2021@163.com; 5School of Life Science, Fudan University, Shanghai 200438, China; 20110700150@fudan.edu.cn; 6College of Veterinary Medicine, Shanxi Agricultural University, Taigu 030801, China

**Keywords:** *Giardia duodenalis*, *Enterocytozoon bieneusi*, *Cryptosporidium* spp., zoonotic genotypes, pet dogs and cats, Yunnan province, China

## Abstract

*Giardia duodenalis*, *Enterocytozoon bieneusi* and *Cryptosporidium* spp. are common enteric pathogens that reside in the intestines of humans and animals. These pathogens have a broad host range and worldwide distribution, but are mostly known for their ability to cause diarrhea. However, very limited information on prevalence and genotypes of *G. duodenalis*, *E. bieneusi* and *Cryptosporidium* spp. in pet dogs and cats are available in China. In the present study, a total of 433 fecal samples were collected from 262 pet dogs and 171 pet cats in Yunnan province, southwestern China, and the prevalence and the genotypes of *G. duodenalis*, *E. bieneusi* and *Cryptosporidium* spp. were investigated by nested PCR amplification and DNA sequencing. The prevalence of *G. duodenalis*, *E. bieneusi* and *Cryptosporidium* spp. was 13.7% (36/262), 8.0% (21/262), and 4.6% (12/262) in dogs, and 1.2% (2/171), 2.3% (4/171) and 0.6% (1/171) in cats, respectively. The different living conditions of dogs is a risk factor that is related with the prevalence of *G. duodenalis* and *E. bieneusi* (*p* < 0.05). However, there were no statistically significant difference in prevalence of three pathogens in cats. DNA sequencing and analyses showed that four *E. bieneusi* genotypes (PtEb IX, CD9, DgEb I and DgEb II), one *Cryptosporidium* spp. (*C. canis*) and two *G. duodenalis* assemblages (C and D) were identified in dogs; two *E. bieneusi* genotypes (Type IV and CtEb I), one *Cryptosporidium* spp. (*C. felis*) and one *G. duodenalis* assemblage (F) were identified in cats. Three novel *E. bieneusi* genotypes (DgEb I, DgEb II and CtEb I) were identified, and the human-pathogenic genotypes/species Type IV *C. canis* and *C. felis* were also observed in this study, indicating a potential zoonotic threat of pet dogs and cats. Our results revealed the prevalence and genetic diversity of *G. duodenalis*, *E. bieneusi* and *Cryptosporidium* spp. infection in pet dogs and cats in Yunnan province, southwestern China, and suggested the potential threat of pet dogs and cats to public health.

## 1. Introduction

*Giardia duodenalis*, *Cryptosporidium* spp. and *Enterocytozoon bieneusi* are three eukaryotic unicellular protozoans, which are the causative pathogens of giardiasis, cryptosporidiosis, and microsporidiosis, respectively [1,2,3,4]. These pathogens can cause many gastrointestinal symptoms such as abdominal pain, nausea, vomiting, anorexia and weight loss especially acute and chronic diarrhea [5,6,7,8,9,10]. Humans and various animals can be infected by *G. duodenalis, Cryptosporidium* spp. and *E. bieneusi* through fecal-oral transmission of their cysts or spores [11,12]. 

At present, eight *G. duodenalis* assemblages (A–H) have been identified by the molecular biological detection method [13]. Among these genotypes, assemblages A and B are regarded as zoonotic assemblages which mainly infect humans and other mammals [14]. Other *G. duodenalis* assemblages (C–H) are commonly considered as host-specific, while assemblages C and D are usually canine-specific assemblages, and assemblage F is usually a feline-specific assemblage [15,16]. However, assemblages E and F have also been detected in humans [17,18]. In total, over 40 *Cryptosporidium* species have been reported, and over 21 species have been reported in humans, including *C. canis* and *C. felis*, which cause the vast majority of infections in dogs and cats, respectively [12,19]. Moreover, *Cryptosporidium muris*, *Cryptosporidium parvum* and *Cryptosporidium ubiquitum* have also been reported in dogs and cats [6,7,20,21,22,23,24]. *Enterocytozoon bieneusi* is the most common species causing human gut infections among nearly 1500 microsporidian species [23]. At least 500 *E. bieneusi* genotypes have been defined thus far, which can be divided into several genetically isolated groups, including zoonotic groups (Group 1 and Group 2) and host adapted groups (Groups 3 to 11) [23,24]. 

Due to the closer relationships between humans with pet dogs and cats, many pathogens can be transmitted to humans through pet dogs and cats, including *G. duodenalis*, *Cryptosporidium* spp. and *E. bieneusi.* Therefore, investigation of the prevalence and genotypes/species of *G. duodenalis*, *Cryptosporidium* spp. and *E. bieneusi* in pet dogs and cats will improve our understanding of the potential threat posed by these pathogens in companion animals in Yunnan province, China.

## 2. Results 

### 2.1. Prevalence of G. duodenalis, E. Bieneusi and Cryptosporidium spp. in Pet Dogs and Cats

The prevalence of *G. duodenalis*, *Cryptosporidium* spp. and *E. bieneusi* was 13.7% (95%CI 9.6–17.9), 4.6% (95%CI 2.0–7.1), 8.0% (95%CI 4.7–11.3) in dogs; and it was 1.2% (95%CI 0–2.8), 0.6% (95%CI 0–1.7) and 2.3% (95%CI 0.1–4.6) in cats, respectively (Table 1). Among three regions, the prevalence of *G. duodenali* in dogs in Kunming city was significantly higher than that in Chuxiong city and Lijiang city (*p* < 0.05). Moreover, the prevalence of *G. duodenalis* in dogs in shelter dogs (27.8%, 20/72, 95%CI 17.4–38.1) was higher than that in pet markets (2.9%, 1/34, 95%CI 0–8.6) and pet hospitals (9.6%, 15/156, 95%CI 5.0–14.2), and the difference was statistically significant (*p* < 0.001). However, no statistically significant difference in prevalence of *G. duodenalis* in pet cats was observed (Table 1). 

Among the different living conditions of dogs, the difference in *E. bieneusi* prevalence was statistically significant (*p* < 0.001). The prevalence of *E. bieneusi* in dogs aged more than 6 months was 10.3% (95%CI 6.0–14.6), which was significantly higher than that in dogs aged less than 6 months (1.5%, 95%CI 0–4.3) (Table 1). Also, the prevalence of *E. bieneusi* in female dogs was 10.3% (95%CI 5.5–15.1), which was higher than that in male dogs (4.7%, 95%CI 0.7–8.7), but the difference in prevalence was not statistically significant (*p* = 0.098). Similarly, the prevalence of *E. bieneusi* in female cats (3.3%, 95%CI 0–7.9) was slightly higher than that in male cats (1.8%, 95%CI 0–4.3) (Table 1). 

Furthermore, the prevalence of *Cryptosporidium* spp. in dogs in shelter (15.3%, 95%CI 7.0–23.6) was higher than that in pet markets (no detection) and pet hospitals (0.6%, 95%CI 0–1.9). Between two gender groups, the prevalence of *Cryptosporidium* spp. in male and female dogs was not significantly different (Table 1). 

### 2.2. Assemblages and Subtypes of G. duodenalis in Pet Dogs and Cats

PCR amplification and DNA sequencing showed that 38 positive samples (36 from dogs and 2 from cats) of *G. duodenalis* were detected at bg locus, resulting three assemblages, namely C (4 from dogs), D (32 from dogs) and F (2 from cats). In addition, at the gdh locus, the 19 gdh-positive samples were identified as assemblage C (4 from dogs), D (13 from dogs) and F (2 from cats). Only one tpi-positive sample (1 from dogs) was identified as assemblage C. 

Sequence alignment analysis revealed some single nucleotide polymorphisms at bg-sequences, gdh-sequences and tpi-sequences, respectively. At bg locus, one subtype of assemblage C, 7 subtypes of assemblage D and one subtype of assemblage F were identified, including five novel (Da4 * ~ Da7 *, Fa1 *) and four known sub-assemblages (Table 2). Also, at gdh gene locus, three subtypes of assemblage C, seven subtypes of assemblage D and one subtype of assemblage F were identified, including four novel (Cb3 *, Db5 * ~ Db7 *) and six known subtypes (Table 2). Only one novel subtype (Cc1 *) of assemblage C was found at tpi gene locus (Table 2). Moreover, one sample were successfully amplified and sequenced at three gene loci (bg, gdh and tpi), forming one mixed infection (Table 3).

### 2.3. Genotypes of Enterocytozoon bieneusi and Cryptosporidium spp. in Pet Dogs and Cats

Based on the ITS sequence, a total of four genotypes, including two known genotypes PtEb IX (*n* = 18), CD9 (*n* = 1) in dogs and two novel genotypes DgEb I (*n* = 1) and DgEb II (*n* = 1) were identified in pet dogs, and one known genotype Type IV (*n* = 3) and one novel genotype CtEb I (*n* = 1) were identified in pet cats (Table 4). The phylogenetic tree showed that genotypes DgEb I, DgEb II, PtEb IX and CD9 all belonged to the dog-specific group. However, genotypes Type IV and CtEb I belonged to the zoonotic Group 1 (Figure 1). Moreover, mixed infections with more than one genotype of *E. bieneusi* in dogs and cats were not detected. 

Two *Cryptosporidium* species were identified among the 13 *Cryptosporidium*-positive samples, including 12 samples of *C. canis* in dogs and one sample of *C. canis* in cats (Table 4). Five nucleotide sequences of *C. canis* obtained in this study had 100% similarity to those deposited sequences in GenBank under accession numbers MN696800. Other sequences of *C. canis* had 99% similarity to those deposited sequences in GenBank under accession number KR999984 and KT749818, respectively (Table 4). Moreover, only one *C. canis* sequence had 97% similarity to those deposited sequences in GenBank under accession number KM977642 (Table 4).

## 3. Discussion

Dogs and cats, as domestic animals, share a common environment with humans and other animals, and can infect them with various unicellular zoonotic pathogens. Thus far, many studies about the infection of *G. duodenalis*, *Cryptosporidium* spp. and *E. bieneusi* in dogs and cats have been recorded worldwide, such as Asia, Europe and Latin America, although only a few have been reported in Africa (Table 5) [6,7,16,20,25,26,27,28,29,30,31,32,33,34,35,36,37,38,39,40,41,42,43,44,45,46,47,48,49,50,51,52,53]. According to the studies in China, the prevalence of *G. duodenalis* ranges from 4.5–26.2% in dogs and 1.9–13.1% in cats [6,7,25,26]; the prevalence of *Cryptosporidium* spp. ranges from 3.1–7.5% in dogs and 5.6–5.8% in cats [6,7,46,47]; and the prevalence of *E. bieneusi* ranges from 6.0–13.9% in dogs and 1.4–11.5% in cats [6,7,33,34,35], respectively (Table 5). 

In the present study, the prevalence of *G. duodenalis* in dogs is higher than that in Heilongjiang (4.5%) [6], Guangdong (10.8%) [25] and Sichuan (11.3%) [26] provinces, China, and is also higher than other zoonotic pathogens in dogs, such as 10.3% for *Babesia canis*, 9.1% for *Anaplasma* spp., 4.5% for *Leishmania infantum*, 1.7% for *Borrelia burgdorferi*, 0.4% for *Ehrlichia* spp. and 1.7% for *Dirofifilaria immitis* in Italy [54], but is lower than that in Henan province (14.3%) [27], Shanghai city (26.2%) in China [7] and other countries (Table 5). Similarly, the *G. duodenalis* prevalence in pet cats is consistent with that in Hangzhou city (1.2%) [28], China; but is lower than that in Heilongjiang (1.9%) [6] and Guangdong (5.8%) provinces [25] and Shanghai city (13.1%) [7] in China and other countries (Table 5), and is also lower than *L. infantum* (3.0%) in Greece and Italy; *Rickettsia felis* (10.8%), *Rickettisa typhi* (4.2%), *Anaplasma phagocytophilum* (2.4%) and *Ehrlichia canis* (2.4%) in cats in Italy [55,56]. The reason is complicated among different studies because many factors could affect the prevalences such as sample sizes, sample sources, environments, animal welfare, hygiene conditions, age and sex of samples, and the sensitivity of tested methods. Moreover, the living condition is a risk factor (*p* < 0.05) that is significantly related to the prevalence of *G. duodenalis* in pet dogs in this study. We suspect that the poor sanitation of shelters contributes significantly to nosocomial transmission, adding to the prevalence of *G. duodenalis* in pet dogs. Furthermore, the higher prevalence of *G. duodenalis* was detected in pet dogs in Kunming city (*p* < 0.05) (Table 1), which suggests that the region is also a risk factor significantly associated with *G. duodenalis* infection in this study. In addition, the prevalence of *G. duodenalis* in male dogs was higher than that in female dogs in the present study, which is consistent with observations in other previous studies [2,57], although the difference was not statistically significant (*p* > 0.05). Compared with dogs, cats seem to be less susceptible to infection with *G. duodenalis* (Table 1). This might be explained by the different living habits of these two animals.

Similar to *G. duodenalis*, the prevalences of *E. bieneusi* in pet dogs and cats in different regions are different (Table 5). This is probably because the route and source of infection for dogs or cats in each region may be different. In addition, other factors can also affect the prevalence of *E. bieneusi* in dogs and cats. Furthermore, statistical analysis showed that a significant difference was observed among pet dogs in shelters, pet markets and pet hospitals (Table 1), which indicates that dogs living in shelters are more easily infected with *E. bieneusi* than those dogs in pet hospitals and markets. The reason may be the poorer hygiene conditions in shelters compared with pet markets and pet hospitals. Pet dogs aged more than 6 mouths seemed to be more susceptible to infection with *E. bieneusi* (*p* < 0.05) (Table 1), suggesting that further relevant research should pay more attention to the adult dogs. Additionally, only cats in Chuxiong city were found to be infected by *E. bieneusi* (Table 1); thus, we speculate that the regional factors may have a significant effect on the prevalence of *E. bieneusi* in cats. But this hypothesis needs to be tested. Additionally, there was no significant difference in the prevalence of *Cryptosporidium* spp. rate in pet dogs or cats (Table 1). 

Up to now, six assemblages (assemblage A, B, C, D, E and F) have been identified in dogs and cats in previous studies [6,7,25,26,27,31], and canine-specific and feline-specific assemblages C, D and F are also found in other animals [11]. These findings indicate that both dogs and cats are a reservoir of *G. duodenalis,* which has risk of transmission among different animals. In the present study, only two assemblages (C and D) were identified in pet dogs, which is similar to previous studies [26,27]. Furthermore, a previous work demonstrated that the assemblages C and D are more sensitive than assemblage A in pet dogs [58]. Moreover, we found nine subtypes of assemblage (at bg locus, *n* = 4, at gdh locus, *n* = 4 and at tpi locus, *n* = 1) in dogs and one subtype of assemblage (at bg locus, *n* = 1) in cats (Table 2). The assemblage of *G. duodenalis* in dogs in the current study seems to more likely to mutate, thus further studies need to examine the genetic structure of these subtypes. Also, one mixed genotype of *G. duodenalis* was found in dogs in this study (Table 3), revealing the diversity of *G. duodenalis* in our investigation area.

Early studies have reported that genotypes of *E. bieneusi* CD1 to CD8, D, O, PigEBITSS, EbpA, CMl, Peru8 and EbpC are identified in dogs, and genotypes D, BEB6, I, CC1, CC2, CC3, CC4 are identified in cats in other provinces of China [33,59]. In the present study, the dominant genotype of *E. bieneusi* PtEb IX (18/21) is a common dog-specific *E. bieneusi* genotype identified in dogs (Table 4). Additionally, two novel genotypes (DgEb I and DgEb II) were also identified in dogs in our study, which enrich the genotype variety of *E. bieneusi* in dogs. *E. bieneusi* genotype Type IV and novel genotype CtEb I in pet cats belonged to Group 1 of zoonotic potential (Figure 1), which imply that pet cats may be a potential source of human infection with *E. bieneusi* in Yunnan province, China.

According to previous studies, *C. ubiquitum* and *C. canis* are commonly found in dogs, and *C. parvum* and *C. felis* are commonly found in cats in Heilongjiang, Shanghai and other cities or provinces of China [6,7]. In the present study, we only identified *C. canis* and *C. felis* in pet dogs and cats, respectively (Table 4). By contrast with the current study, the *C. parvum* and *C. muris* have been found in dogs or cats in other countries [20,21,22,36,60]. Despite our results revealing the presence of host-specific *Cryptosporidium* spp. species (*C. canis* and *C. felis*) in pet dogs and cats, these two species have been reported in humans and mainly in developing countries [6]. This finding suggests that people still need to take further precautions when they are in close contact with their pets. In addition, some nucleotide sequences of *Cryptosporidium* spp. obtained in pet dogs and cats have mutations in this study (Table 4). 

## 4. Materials and Methods 

### 4.1. Study Sites

The fecal samples of pet dogs and cats were collected in Kunming city, Lijiang city and Chuxiong city in Yunnan province (Location: 21°8′ N to 29°15′ N and 97°31′ E to 106°11′ E), southwestern China, which covers more than 390,000 square kilometers and has a population of approximately 48 million. 

### 4.2. Sampling

During August to September 2018, a total of 433 fresh fecal samples were collected from pet dogs and cats in three cities of Yunnan province, including Kunming city (134 dogs and 36 cats), Lijiang city (90 dogs and 110 cats) and Chuxiong city (38 dogs and 25 cats). The Kunming, Lijiang and Chuxiong cities have more numbers of pet dogs and cats than other cities of Yunnan province, and all the samples of the cats and dogs were randomly collected from the biggest pet hospital, pet market and shelter in each city (i.e., Kunming city, Lijiang city and Chuxiong city), respectively. Moreover, the information regarding regions, ages, genders and living conditions were recorded. All the fecal samples were saved into 15 mL centrifuge tube with 2.5% potassium dichromate, and then were stored at 4 °C until for DNA extraction.

### 4.3. Genomic DNA Extraction and PCR Amplification 

Each fecal sample was washed three times with distilled water by centrifuging at 13,000 *g* for 5 min to remove potassium dichromate, and 300 mg of the precipitated samples were used for DNA extraction using the E.Z.N.A. Stool DNA kit (OMEGA, Biotek Inc. USA) according to the manufacturer’s instructions. The genomic DNA was stored at –20 °C before PCR amplification. The *G. duodenalis* identification was performed by nested PCR amplification of bg, gdh and tpi gene loci according to previous reports [25,61], *Cryptosporidium* spp. identification was conducted by nested PCR amplification of the 18S ribosomal RNA [62], and *E. bieneusi* identification was carried out by nested PCR amplification of ITS rDNA sequences as previously described [63]. The positive and negative controls were included in each PCR reaction. All the secondary PCR products were checked by 2% (*w*/*v*) agarose gel electrophoresis after ethidium bromide staining and visualized under UV light.

### 4.4. Sequence Analysis

The PCR-positive products were sent to Tsingke Biological Technology Company (Xi’an, China) for two-directional sequencing. The obtained sequences were spliced together after initial collation with their DNA peak form graph by Chromas v.2.6. The genotypes/species of *G. duodenalis*, *Cryptosporidium* spp. and *E. bieneusi* were identified by aligning the obtained sequences with corresponding sequences in the GenBank database (http://www.ncbi.lm.nih.gov/GenBank/, accessed on 11 July 2021). The phylogenetic tree was established by neighbor-joining method (NJ) with Kimura 2-parameter model in MEGA 7.0 (http://www.megasoftware.net/, accessed on 11 July 2021). The novel genotypes of *E. bieneusi* were decided by the ~243-bp ITS region [64,65].

### 4.5. Statistical Analysis

Prevalence of *G. duodenalis*, *Cryptosporidium* spp. and *E. bieneusi* in age, regio, gender and living conditions groups were analyzed using Chi-square test in SPSS 24.0 (SPSS Inc., Chicago, IL, USA). The 95% confidence intervals (CIs) were estimated. The difference was considered statistically significant when *p*-value < 0.05. 

## 5. Conclusions

The present investigation revealed the prevalence and assemblages/genotypes/species of *G. duodenalis*, *E. bieneusi* and *Cryptosporidium* spp. in pet dogs and cats in Yunnan province, China. The infection with *G. duodenalis*, *E. bieneusi* and *Cryptosporidium* spp. in dogs and cats suggests that we should take measures to prevent and control those pathogens from being transmitted to other animals and humans. Our data provided the valuable information for a better understanding of the epidemiology and public health threat of *Giardiasis, E. bieneusi* and *Cryptosporidium* spp. in pet dogs and cats in southwestern China.

## Figures and Tables

**Figure 1 pathogens-10-01107-f001:**
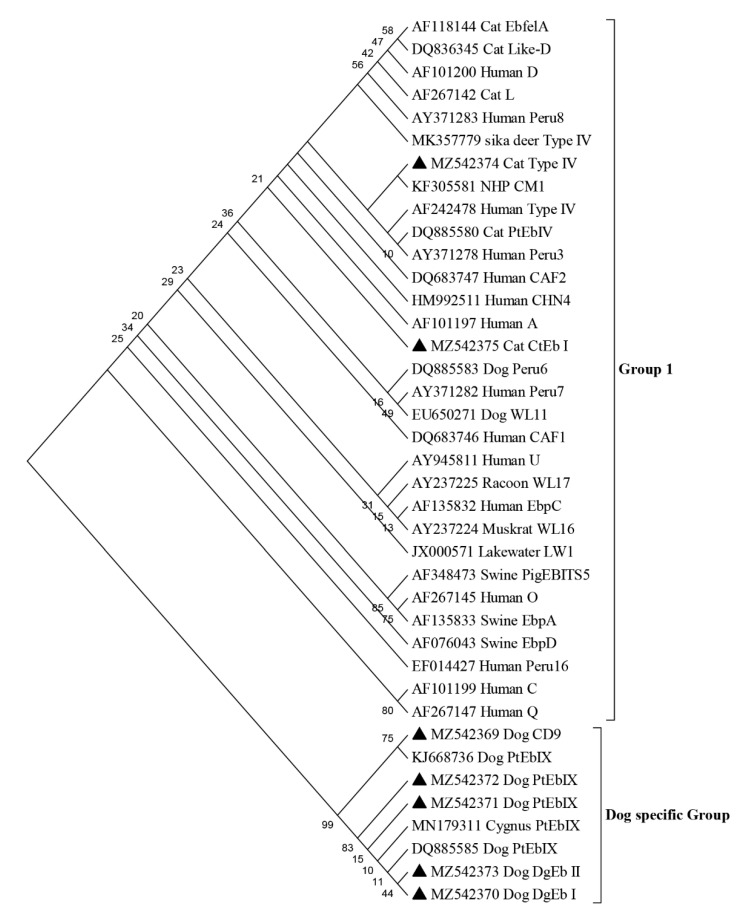
Phylogenetic relationship based on ITS sequences of *Enterocytozoon bieneusi* in pet dogs and cats in Yunnan province, southwestern China. (Note: The samples in this study are indicated by triangles).

**Table 1 pathogens-10-01107-t001:** Factors associated with *Giardia*
*duodenalis*, *Enterocytozoon bieneusi* and *Cryptosporidium* spp. prevalence in pet dogs and cats in Yunnan province, southwestern China.

Animals	Factors	Category	No. Sample	*Giardia duodenalis*			*Enterocytozoon bieneusi*			*Cryptosporidium* spp.		
				No. Positive	% (95% CI)	*p*-Value	No. Positive	% (95% CI)	*p*-Value	No. Positive	% (95% CI)	*p*-Value
Dogs	Age	<6 months	68	8	11.8 (4.1–19.4)	0.582	1	1.5 (0–4.3)	0.021	1	1.5 (0–4.3)	0.154
		>6 months	194	28	14.4 (9.5–19.4)		20	10.3 (6.0–14.6)		11	5.7 (2.4–8.9)	
	Region	Kunming	134	26	19.4 (12.7–26.1)	0.013	18	13.4 (7.7–19.2)	-	11	8.2 (3.6–12.9)	-
		Lijiang	90	9	10.0 (3.8–16.2)		0	0		1	1.1 (0–3.3)	
		Chuxiong	38	1	2.6 (0–7.7)		3	7.9 (0–16.5)		0	0	
	Gender	Male	107	16	15.0 (8.2–21.7)	0.636	5	4.7 (0.7–8.7)	0.098	5	4.7 (0.7–8.7)	0.95
		Female	155	20	12.9 (7.6–18.2)		16	10.3 (5.5–15.1)		7	4.5 (1.2–7.8)	
	Living condition	Pet hospital	156	15	9.6 (5.0–14.2)	<0.001	1	0.6 (0–1.9)	<0.001	1	0.6 (0–1.9)	-
		Pet market	34	1	2.9 (0–8.6)		3	8.8 (0–18.4)		0	0	
		Shelter	72	20	27.8 (17.4–38.1)		17	23.6 (13.8–33.4)		11	15.3 (7.0–23.6)	
		Subtotal	262	36	13.7 (9.6–17.9)		21	8.0 (4.7–11.3)		12	4.6 (2.0–7.1)	
Cats	Age	< 6 months	145	2	1.4 (0–3.3)	-	4	2.8 (0.1–5.4)	-	1	0.7 (0–2.0)	-
		> 6 months	26	0	0		0	0		0	0	
	Region	Kunming	36	1	2.8 (13.1–42.4)	-	0	0	-	0	0	-
		Lijiang	110	0	0		0	0		1	0.9 (0–2.7)	
		Chuxiong	25	1	4.0 (0–11.7)		4	16.0 (1.6–30.4)		0	0	
	Gender	Male	111	2	1.8 (0–4.3)	-	2	1.8 (0–4.3)	-	1	0.9 (0–2.7)	-
		Female	60	0	0		2	3.3 (0–7.9)		0	0	
	Living condition	Pet hospital	154	2	1.3 (0–3.1)	-	4	2.6 (0.1–5.1)	-	1	2.6 (0–1.9)	-
		Shelter	17	0	0		0	0		0	0	
		Subtotal	171	2	1.2 (0–2.8)		4	2.3 (0.1–4.6)		1	0.6 (0–1.7)	
Total			433	38	8.8 (6.1–11.4)		25	5.8 (3.6–8.0)		13	3.0 (1.4–4.6)	

**Table 2 pathogens-10-01107-t002:** Variations in nucleotide sequences of assemblages of *Giardia duodenalis* in pet dogs and cats in Yunnan province, southwestern China.

Locus	Host (Subtypes)	Nucleotide at Position						No. Positive	Accession Number
bg	**(a) Variations in bg nucleotide sequences among assemblage D**								
		31	61	103	109	203			
	Reference sequences	G	A	G	C	A			MG873354
	Dog (Da1)							20	MN734349
	Dog (Da2)			A				5	MN734350
	Dog (Da3)				T			3	MN734353
	Dog (Da4 *)			A	T			1	MN734351
	Dog (Da5 *)	A			A			1	MN734354
	Dog (Da6 *)		C	A				1	MN734352
	Dog (Da7 *)	A			A	G		1	MN734355
	**(b) Variations in bg nucleotide sequences among assemblage F**								
		55							
	Reference sequences	C							KX960131
	Cat (Fa1 *)	T						2	MN734356
	**(c) Variations in bg nucleotide sequences among assemblage C**								
	Reference sequences								KY979502
	Dog (Ca1)							4	MN734348
gdh	**(a) Variations in gdh nucleotide sequences among assemblage C**								
		340	589	600	603	693			
	Reference sequences	A	G	C	A	G			MF990016
	Dog (Cb1)	G	A	T	G			2	MN734358
	Dog (Cb2)	G	A	T	G	T		1	MN734359
	Dog (Cb3 *)				G			1	MN734357
	**(b) Variations in gdh nucleotide sequences among assemblage D**								
		356	368	386	506	509	654		
	Reference sequences	C	A	T	A	C	A		MF990017
	Dog (Db1)							1	MN734366
	Dog (Db2)	T	G					3	MN734362
	Dog (Db3)			A			G	5	MN734364
	Dog (Db4)	T						1	MN734363
	Dog (Db5*)		G					1	MN734361
	Dog (Db6*)			A				1	MN734360
	Dog (Db7*)	T	G		T	T		1	MN734365
	**(c) Variations in gdh nucleotide sequences among assemblage F**								
	Reference sequences								KM977649
	Cat (Fb1)							2	MN734367
tpi	**Variations in tpi nucleotide sequences among assemblage C**								
		135	315						
	Reference sequences	G	T						KY979494
	Dog (Cc1 *)	T	C					1	MN734368

* means novel subtypes of assemblage.

**Table 3 pathogens-10-01107-t003:** Multilocus characterization of *Giardia duodenalis* isolates based on the bg, tpi and gdh genes.

Isolate	Assemblage			No. Sequences	MLG Type
	bg	tpi	gdh		
XSQG34	D	C	C	1	Mixed

**Table 4 pathogens-10-01107-t004:** Species or genotypes of *Cryptosporidium* spp. and *Enterocytozoon bieneusi* in pet dogs and cats in Yunnan province, southwestern China.

Hosts	*Enterocytozoon bieneusi* Genotype (No.)	GenBank Accession Number in This Study	
Dog	DgEb I * (1)	MZ542370	
Dog	CD9 (1)	MZ542369	
Dog	DgEb II * (1)	MZ542373	
Dog	PtEb IX (1)	MZ542371	
Dog	PtEb IX (17)	MZ542372	
Cat	Type IV (3)	MZ542374	
Cat	CtEb I * (1)	MZ542375	
**Hosts**	***Cryptosporidium* spp. Genotype (No.)**	**Reference Sequences GenBank** **Accession Number**	**Similarity**
Dog	*C. canis* (5)	MN696800	100%
Dog	*C. canis* (4)	KR999984	99%
Dog	*C. canis* (3)	KT749818	99%
Cat	*C. felis* (1)	KM977642	97%

Note: * represent novel genotype.

**Table 5 pathogens-10-01107-t005:** Prevalence of *Giardia*
*duodenalis, Enterocytozoon bieneusi* and *Cryptosporidium* spp. in dogs and cats in different regions of the world.

Regions	Hosts	Prevalence (%)	Hosts	Prevalence (%)	Reference
**(a) Prevalence of *Giardia duodenalis* in dogs and cats in different regions of the world.**
China					
Shanghai	Dogs	26.2%	Cats	13.1%	[7]
Guangdong	Dogs	10.8%	Cats	5.8%	[25]
Heilongjiang	Dogs	4.5%	Cats	1.9%	[6]
Sichuan	Dogs	11.3%	-	-	[26]
Henan	Dogs	14.3%	-	-	[27]
Hangzhou	-	-	Cats	1.2%	[28]
Yunnan	Dogs	13.7%	Cats	1.2%	Present study
Other countries					
Australia	Dogs	6.3%	Cats	2.0%	[20]
Greece	Dogs	25.2%	Cats	20.5%	[29]
Spain	Dogs	33%	Cats	9.2%	[30]
Ontario	Dogs	64.0%	Cats	87.0%	[31]
Brazil	Dogs	19.6%	-	-	[32]
**(b) Prevalence of *Enterocytozoon bieneusi* in dogs and cats in different regions of the world.**
China					
Shanghai	Dogs	6.0%	Cats	5.6%	[7]
Heilongjiang	Dogs	6.7%	Cats	5.8%	[6]
Henan	Dogs	13.9%	Cats	11.5%	[33]
Eastern China	Dogs	8.6%	Cats	1.4%	[34]
Changchun	Dogs	7.8%	-	-	[35]
Yunnan	Dogs	8.0%	Cats	2.3%	Present study
Other countries					
Colombia	Dogs	15.0%	Cats	17.4%	[36,37]
Egypt	Dogs	13.0%	Cats	12.5%	[38]
Germany	Dogs	0.0%	Cats	5.0%	[39]
Spain	Dogs	0.8%	Cats	3.0%	[40]
Japan	Dogs	2.5%	Cats	14.3%	[41]
Poland	Dogs	4.9%	Cats	9.1%	[42]
Thailand	Dogs	0.0%	Cats	31.3%	[43]
Portugal	Dogs	100.0%	Cats	100.0%	[44]
Iran	Dogs	25.8%	Cats	7.5%	[45]
**(c) Prevalence of *Cryptosporidium* spp. in dogs and cats in different regions of the world.**
China					
Shanghai	Dogs	6.0%	Cats	5.6%	[7]
Heilongjiang	Dogs	6.7%	Cats	5.8%	[6]
Zhengzhou	Dogs	3.1%	-	-	[46]
Ya’an	Dogs	7.5%	-	-	[47]
Yunnan	Dogs	4.6%	Cats	0.6%	Present study
Other countries					
Japan	-	-	Cats	1.4%	[48]
Spain	Dogs	5.5%	Cats	8.8%	[16]
Germany	Dogs	1.2%	Cats	5.3%	[49]
Greece	Dogs	5.9%	Cats	6.8%	[29]
Thailand	Dogs	2.1%	Cats	2.5%	[50]
Brasil	Dogs	24.5%	Cats	11.1%	[51]
Italy	Dogs	1.7%	-	-	[52]
Netherland	Dogs	8.7%	Cats	4.6%	[53]

## Data Availability

The data that support the figures within this paper and other finding of this study are available from the corresponding authors upon reasonable request. All of the obtained representative *G. duodenalis* bg, gdh and tpi nucleotide sequences were deposited in GenBank (https://www.ncbi.nlm.nih.gov/ accessed on 26 November 2019) under the accession numbers MN734348- MN734356, MN734357-MN734367 and MN734368, respectively. The nucleotide sequences of *Cryptosporidium* spp. and *E. bieneusi* were deposited in GenBank under accession numbers MZ540366-MZ540371 and MZ542369-MZ542375, respectively.

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
