# Peer review of "Molecular Investigation of Zoonotic Intestinal Protozoa in Pet Dogs and Cats in Yunnan Province, Southwestern China"

_pathogens, 2021, doi:10.3390/pathogens10091107_

Round 1
Reviewer 1 Report
The manuscript on the molecular investigation of zoonotic intestinal protozoa in pet dogs and cats in Yunnan Province in China attempted to clarify the prevalence and genotypes of three common enteric pathogens in pet dogs and cats. In their study, 433 fecal samples (262 from pet dogs and 171 from pet cats) were collected and analysed for prevalence and genotypes of the protozoa. The authors report a three novel E. bieneusi genotypes and highlight a potential zoonotic threat of pet dogs and cats. I found the manuscript to be well written but also place the results in the global context. I have some few questions for the authors though:
i. From Table 1 - What was the criteria for the selection of the three regional study areas; Kumming, Lijiang and Chuxion? and ii. how was the sampling of the dogs and cats in the study done?
iii. I suggest that the authors highlight in their discussion the scarcity of data from other parts of the world particularly Africa and Asia. If one looks at Table 5, only Egypt (from Africa) and no other country from Asia is mentioned.
iv. Page 7 line 115 - How do the authors describe a reservoir for G. duodenalis?
Author Response
Responses to comments and suggestions of Reviewer #1:
General comments:
The manuscript on the molecular investigation of zoonotic intestinal protozoa in pet dogs and cats in Yunnan Province in China attempted to clarify the prevalence and genotypes of three common enteric pathogens in pet dogs and cats. In their study, 433 fecal samples (262 from pet dogs and 171 from pet cats) were collected and analysed for prevalence and genotypes of the protozoa. The authors report three novel E. bieneusi genotypes and highlight a potential zoonotic threat of pet dogs and cats. I found the manuscript to be well written but also place the results in the global context. I have some questions for the authors though.
Response: We thank Reviewer #1 very much for favorable comments and suggestions on MS.
Minor point:
Q1:From Table 1 - What was the criteria for the selection of the three regional study areas; Kumming, Lijiang and Chuxiong?
Response: We thank Reviewer #1 very much for the query. Kunming, Lijiang and Chuxiong cities have more numbers of pet dogs and cats than other cities of Yunnan province, and the results are good representatives for the prevalence of Giardia duodenalis, Enterocytozoon bieneusi and Cryptosporidium spp. in pet dogs and cats in Yunnan province, southwestern China. We have added all these information into the Methods Section in the revised MS.
Q2: How was the sampling of the dogs and cats in the study done?
Response: We thank Reviewer #1 very much for the query. All the samples of the animals were randomly collected from the biggest pet hospital, pet market and shelter in each city (ie., Kunming city, Lijiang city and Chuxiong city), respectively. We collected fresh faeces from pet dogs and cats, and saved into 15 ml centrifuge tube with 2.5% potassium dichromate, and then were stored at 4 °C until for DNA extraction. We have added all these information into the Methods Section in the revised MS.
Q3: I suggest that the authors highlight in their discussion the scarcity of data from other parts of the world particularly Africa and Asia. If one looks at Table 5, only Egypt (from Africa) and no other country from Asia is mentioned.
Response: We thank Reviewer #1 very much for the suggestion. We have highlighted the scarcity of data from other parts of the world particularly Africa and Asia in the discussion accordingly.
Q4: Page 7 line 115 - How do the authors describe a reservoir for G. duodenalis?
Response: We think pet dogs and cats have close contact with humans and other animals. Thus, once being infected with zoonotic pathogens, the pet dogs and cats may serve as reservoir that would potentially transmit the pathogens to humans and other different animals.
We sincerely hope that you find our MS revised to your satisfaction. We are looking forward to receiving your editorial decision soon and hope to see our work published in Pathogens.
With best wishes,
Yang Zou,
On behalf of all co-authors.
Reviewer 2 Report
This ms describes a Study aiming at investigating the occurrence of zoonotic protozoa in pet dogs and cats in a region of China. Minor revisions are necessary before considering the ms suitable for publication.
-Please check the ms for misspellings, consistency and typos (e.g. spp. should not be in italics, and the suffix -iasis is only for the infection in humans, while the suffix -osis should be used for the infection in animals).
-The discussion could be implemented with epidemiological comparisons with other unicellular zoonotic pathogens of dogs and cats, see for instance:
https://pubmed.ncbi.nlm.nih.gov/33922459/
https://pubmed.ncbi.nlm.nih.gov/33363245/
https://pubmed.ncbi.nlm.nih.gov/32441628/
https://pubmed.ncbi.nlm.nih.gov/31437677/
https://pubmed.ncbi.nlm.nih.gov/33032588/
https://pubmed.ncbi.nlm.nih.gov/30917860/
https://pubmed.ncbi.nlm.nih.gov/33363242/
https://pubmed.ncbi.nlm.nih.gov/33802644/
Author Response
Responses to comments and suggestions of Reviewer #2:
General comments:
This MS describes a study aiming at investigating the occurrence of zoonotic protozoa in pet dogs and cats in a region of China. Minor revisions are necessary before considering the MS suitable for publication.
Response: We thank Reviewer #2 very much for favorable comments and suggestions on our MS.
Major points:
Q1: Please check the MS for misspellings, consistency and typos (e.g. spp. should not be in italics, and the suffix -iasis is only for the infection in humans, while the suffix -osis should be used for the infection in animals).
Response: We thank Reviewer #2 very much for excellent comments and suggestions. We have corrected these mistakes in the main text of the MS. Giardiasis can infection in humans and animals, and we found that the word ‘Giardiasis’ is used in almost studies.
Q2: The discussion could be implemented with epidemiological comparisons with other unicellular zoonotic pathogens of dogs and cats, see for instance:
https://pubmed.ncbi.nlm.nih.gov/33922459/
https://pubmed.ncbi.nlm.nih.gov/33363245/
https://pubmed.ncbi.nlm.nih.gov/32441628/
https://pubmed.ncbi.nlm.nih.gov/31437677/
https://pubmed.ncbi.nlm.nih.gov/33032588/
https://pubmed.ncbi.nlm.nih.gov/30917860/
https://pubmed.ncbi.nlm.nih.gov/33363242/
https://pubmed.ncbi.nlm.nih.gov/33802644/
Response: We thank Reviewer #2 very much for constructive suggestions. We have added the epidemiological comparisons with other unicellular zoonotic pathogens of dogs and cats in the discussion.
We have done our best to address all comments and we sincerely hope that you find our MS revised to your satisfaction. We are looking forward to receiving your editorial decision soon and hope to see our work published in Pathogens.
With best wishes,
Yang Zou
On behalf of all co-authors.